# Enhancing the Biocontrol Potential of the Entomopathogenic Fungus in Multiple Respects via the Overexpression of a Transcription Factor Gene *MaSom1*

**DOI:** 10.3390/jof8020105

**Published:** 2022-01-21

**Authors:** Yanru Du, Yuxian Xia, Kai Jin

**Affiliations:** 1School of Life Sciences, Chongqing University, Chongqing 401331, China; 20152602008@cqu.edu.cn; 2Chongqing Engineering Research Center for Fungal Insecticide, Chongqing 401331, China; 3Key Laboratory of Gene Function and Regulation Technologies under Chongqing Municipal Education Commission, Chongqing 401331, China

**Keywords:** *Metarhizium acridum*, *MaSom1* overexpression, stress tolerance, conidiation, virulence

## Abstract

Entomopathogenic fungi play important roles in the control of populations of agricultural and disease vector pests in nature. The shortcomings of mycoinsecticides for pest management in the field cannot be completely overcome by improving single biocontrol properties of fungi. Therefore, enhancing the biocontrol potential of entomopathogenic fungi in multiple respects by genetic engineering is desirable. Transcription factors are usually involved in various important processes during fungal growth and pathogenesis via regulating a series of genes, and are important candidates for fungal improvement via genetic engineering. Herein, overexpression of *MaSom1*, a key transcription factor gene in the cAMP/PKA pathway, improves the biocontrol traits of *Metarhizium acridum* in multiple respects. When compared with WT, the *MaSom1*-overexpression strains exhibit enhanced tolerances to UV-B and heat shock, with increased mean 50% inhibition times by 66.9% and 155.2%, respectively. Advanced conidiation emerged accompanied by increased conidial yield up to 3.89 times after 3-day incubation for the *MaSom1*-overexpression strains compared to WT. Furthermore, when compared with WT, the virulence of the *MaSom1*-overexpression strains was also increased with the mean 50% lethality times reduced by 21.8% to 23.8%. Taken together, the *MaSom1*-overexpression improved the biocontrol potential of *M. acridum* in multiple respects. Our results provide insights into the application of key transcription factors for genetic engineering and offer a credible way to further improve the biocontrol potential of entomopathogenic fungi.

## 1. Introduction

As disease-causing vectors or agricultural pests, insect pests cause huge human health problems and significant economic losses worldwide [1]. The desire for outing the dilemma fuels our interest in various pest control mechanisms, including using insect microbial pathogens [2,3,4]. As the largest group of insect microbial pathogens, entomopathogenic fungi undertake the main missions for insect control in nature, and can infect by penetrating the host cuticle. They show great biocontrol potential, along with the advantages of environmental friendliness, safety to human and animals, and their being easily epidemic in the insect pest population [5,6]. However, some disadvantages, such as poor efficacy, high cost for producing, and susceptibilities to various adverse conditions, have delayed their wide application [5,7,8,9].

Commercial applications require entomopathogenic fungi to kill the host quickly and be able to maintain functions in diverse environmental conditions [4]. Therefore, the screening or improvement of fungal strains with high biocontrol potential is urgently needed. Along with the development of biological sequencing and robust technologies for fungal transformation, genetic engineering makes it possible to improve the mycoinsecticides’ efficacy and cost-effectiveness [10,11].

Abiotic stresses, such as ultraviolet radiation, high temperature or oxidative stress, are key factors for restricting the large-scale application of entomopathogenic fungi in the field [12,13,14]. A series of attempts to decrease the abiotic stress susceptibilities have been performed by genetic engineering. Cotransforming three melanin biosynthesis genes, including a 1,3,8-trihydroxynapthalnene reductase gene, a scytalone dehydratase gene and a polyketide synthase gene from *Alternaria alternata* into *Metarhizium anisopliae* enhanced the fungal tolerances to ultraviolet irradiation and heat shock [15]. Transgenic entomopathogenic fungi with a highly efficient archaeal photolyase exhibited enhanced UV tolerance [16]. Overexpression of heat shock protein gene (*Hsp25*) in *M. roberstsii*, not only increased the tolerance to osmotic stress, but also conferred increased fungal germination [17].

Conidia are the major infectious units of insect pathogenic fungi as well as the main active components of mycoinsecticides, thus the ability to produce conidia is an important aspect for cost effectiveness [18,19]. Clues related to promoting conidiation have been found by genetic engineering. Disruption of *Mamhk1* (a group III histidine kinase gene) resulted in increases in conidial yield and thermotolerance, but decreases in virulence and UV-B tolerance of *M. roberstsii* [20]. Losing *MaPpt1* (a protein phosphatase gene) in *M. acridum* not only enhanced the fungal tolerance to UV-B irradiation but also increased the conidial yield [21].

Virulence in an important criterion for the efficiency of mycoinsecticides. Accumulating data show that genetic engineering has increased fungal virulence via various strategies by exploring clues in both insects and fungi [2]. Firstly, some important genes during fungal pathogenesis were overexpressed in entomopathogenic fungi, accelerating fungal growth in insect hemolymphs or cuticle penetration, such as the acid trehalase gene *MaAtm1* [22], the cuticle-degrading protease gene *MaPr1A* [23] and the chitinase gene *Bbchit1* [10]. Secondly, some insect toxin genes were overexpressed in entomopathogenic fungi, boosting the insecticidal effect of fungi, such as the scorpion neurotoxin genes *AaIT* from *Androctonus australi* [24], *LqhIT2* from *Leiurus quinquestriatus hebraeus* [25], and *BjαIT* from *Buthotus judaicus* [26]. In addition, some insect molecule genes were overexpressed by entomopathogenic fungi, disturbing the development and behaviors of insects, such as the salivary gland and midgut peptide 1 (*SM1*) gene [27], the trypsin-modulating oostatic factor (*Aea*-TMOF) gene [28], and insect serpin gene *Spn43Ac* [29].

However, all of the above fungal improvements are only related to single or partial biocontrol properties of insect pathogenic fungi. The increased conidial yield usually accompanied the achievement of decreased virulence. Upon disruption of *Mamhk1* in *M. roberstsii*, the conidial yield was increased (up to 50%), but the fungal virulence to *Tenebrio molitor larvae* was reduced [2,20]. Increased stress tolerances and/or virulence usually generated impaired conidiation—when a regulated cuticle-degrading protease (*Pr1*) gene was inserted into *M. anisopliae*, the engineered fungus exhibited 25% and 40% reductions in death time and food consumption when compared with wild-type, but the transformants supported little sporulation [23]. In fact, fungal stress tolerances, conidiation and virulence are all key factors to determining the cost-effectiveness and efficacy of mycoinsecticides during field applications. The improvements of single or partial biocontrol properties are not able to overcome the shortcomings of insecticidal fungal pesticides, therefore, enhancing the biocontrol potential of entomopathogenic fungi in multiple respects by genetic engineering is desirable.

Transcription factors may regulate the expression of genes related to various important traits, and could be treated as candidates via genetic engineering for comprehensively enhancing the biocontrol potential of entomopathogenic fungi [30]. In *Saccharomyces cerevisiae*, constitutive expression of the transcription factor gene *Znf1* improved the production of bioethanol and enhanced the fungal tolerance to acetic acid [31]. Som1, a key transcription factor in the cAMP/PKA pathway, contributed to conidiation and virulence in *Magnaporthe oryzae* [32], and regulated fungal growth, conidiation, and virulence in *Aspergillus fumigatus* [33]. Our previous research exhibited that the disruption of *MaSom1* not only affected the conidial yield and stress tolerances seriously, but also decreased the fungal virulence via the impairment of appressorium formation, hyphal bodies’ growth and the ability of avoiding host immune recognition in *M. acridum* [34]. Therefore, overexpression of the transcription factor gene *MaSom1* may improve the biocontrol traits of entomopathogenic fungi in multiple respects due to its critical roles in fungal growth, stress tolerance, conidiation and pathogenicity.

In this work, three *MaSom1*-overexpression strains of *M. acridum* with improved traits in multiple respects were screened from 63 putative *MaSom1*-overexpression transformants and verified via PCR and green fluorescence detection. When compared with WT (the wild-type strain) and VT (*M. acridum* transformed with the pK2-P*_gpdA_*-egfp vector), the *MaSom1*-overexpression strains exhibited enhanced UV-B and heat shock tolerances with significantly lower mean 50% inhibition times (IT_50_s), accelerated conidiation, and promoted appressorium formation. The mean 50% lethality times (LT_50_s) of the *MaSom1*-overexpression strains were significantly decreased after topical inoculation. Taken together, the *MaSom1*-overexpression strains obtain diverse trait improvements for stress tolerance, conidiation and virulence. Our results provide insights to improve the biocontrol potential of entomopathogenic fungi in multiple respects.

## 2. Results

### 2.1. Overexpression of MaSom1 in M. acridum

For genetic improvement of *M. acridum, MaSom1* (GenBank accession No. MAC_02477) was cloned and controlled by the promoter of *gpdA* (the glyceraldehyde-3-phosphate dehydrogenase from *Aspergillus nidulans*, GenBank accession No. Z32524.1) to generate pK2-P*_gpdA_*-MaSom1::egfp vector (Figure 1A) for fungal transformation. After that, three *MaSom1*-overexpression strains which exhibited higher fluorescence in conidia, designated as OE-4, OE-34 and OE-55 from 63 putative *MaSom1*-overexpression transformants, were selected and verified. The quantitative reverse transcription PCR (qRT-PCR) showed that the expression levels of *MaSom1* at conidial stage were increased nearly about 2.59 ± 1.21-fold for OE-4, 2.84 ± 0.83-fold for OE-34, and 2.04 ± 1.11-fold for OE-55 when compared with WT (Figure 1B). Southern blotting was performed to further confirm the *MaSom1* expression strains, and the hybridizing bands for *sur* gene appeared at nearly 5.1 kb, 5.2 kb, 3.0 kb and 5.5 kb for OE-4, OE-34, OE-55 and VT, respectively (Figure 1C). The merged image for fluorescence microscopic observation of egfp and Hoechst 33258 staining of all the fungal strains showed that the gfp fluorescence of OE-4, OE-34 and OE-55 was localized to the nucleus in conidia and hyphae. However, the VT strain, which was transformed with the pK2-P*_gpdA_*-egfp vector, accumulated gfp fluorescence in the cytoplasm of conidia and hyphae (Figure 1D).

### 2.2. Overexpression of MaSom1 Enhanced the Stress Tolerances of M. acridum

Various stresses are the main inhibitors of fungal growth in a natural environment, and the tolerances to heat shock and UV-B of the three *MaSom1*-overexpression strains (OE-4, OE-34 and OE-55) were measured. As shown in Figure 2, the enhanced tolerances to heat shock and UV-B were observed in OE-4, OE-34 and OE-55 when compared with the control strains (WT and VT). After treatment at 45 °C for 8 h, the conidial germination rates reached nearly 29.25% for WT and 44.19% for VT, which were significantly lower than those of OE-4 (64.25%), OE-34 (69.72%) and OE-55 (66.46%, *p* < 0.05; Figure 2A). The IT_50_s of the WT (5.77 ± 1.91 h) and VT (7.66 ± 1.64 h) strains were significantly lower than those of the OE-4 (13.28 ± 7.12 h), OE-34 (14.72 ± 7.16 h) and OE-55 (13.06 ± 4.85 h, *p* < 0.05; Figure 2B). After being treated with UV-B irradiation for 6 h, the conidial germination rates were about 9.33% for WT and 16.17% for VT, which were significantly lower than those of OE-4 (24.22%), OE-34 (45.25%) and OE-55 (30.41%, *p* < 0.05; Figure 2C). The IT_50_s of WT (3.16 ± 0.52 h) and VT (3.80 ± 0.47 h) were significantly lower than those of the OE-34 (5.27 ± 1.05 h) and OE-55 (4.70 ± 0.63 h, *p <* 0.05; Figure 2D). These results indicated that overexpression of *MaSom1* significantly increased the fungal tolerances to heat shock and UV-B irradiation in *M. acridum.* The growth of all the fungal strains were inhibited when calcofluor white (CFW) or Congo red (CR) was added to 1/4 SDAY medium (Figure 2E). The relative growth inhibition of WT and VT was significantly greater than OE-4 and OE-34 when cultured on 1/4 SDAY medium supplemented with CFW (13.35 ± 1.13% for WT, 9.88 ± 1.71% for VT, 5.50 ± 1.93% for OE-4 and 37.51 ± 2.49% for OE-34) and CR (53.72 ± 3.06% for WT, 48.29 ± 4.37% for VT, 40.34 ± 3.01% for OE-4 and 37.51 ± 2.49% for OE-34) (Figure 2F). In addition, qRT-PCR revealed that the expression levels of *MaUve1*, *MaUbi1*, and *MaSsb1*, which are involved in protection against UV-B and heat shock in fungi, were more expressed in *MaSom1*-overexpression strains than that in WT and VT (Figure 2G).

### 2.3. Overexpression of MaSom1 Promoted the Conidiation of M. acridum

As shown in Figure 3, overexpression of *MaSom1* could accelerate the conidiation in *M. acridum*. Morphological observation showed that the conidial germination was advanced in OE-4, OE-34 and OE-55 at 12 h. After inoculation on 1/4 SDAY, the *MaSom1*-overexpression strains formed conidiophores at 18 h, while the control strains did not form conidiophores until 21 h after inoculation (Figure 3B). When compared with WT, the advanced conidiation lead to approximately 2.85, 3.62 and 3.89 times more conidial yield for OE-4, OE-34 and OE-55 after 3 days of incubation, respectively (Figure 3A). Additionally, the conidial yield of OE-4, OE-34 and OE-55 were more dramatically increased than the control strains before incubation for 9 days. However, at the later stage of fungal growth, there were no differences in conidial yield between the *MaSom1*-overexpression strains and WT (Figure 3B).

### 2.4. Overexpression of MaSom1 Augmented the Virulence of M. acridum

Virulence is one of the most important indexes for the entomopathogenic fungi. The bioassays were performed to examine the virulence of the *MaSom1*-overexpression strains. As shown in Figure 4, there is no significant difference in the survival rates of locusts infected by WT and VT strains, but the *MaSom1*-overexpression strains (OE-4, OE-34 and OE-55) exhibited the increased virulence when compared with WT and VT strains. The survival rates of locusts infected by OE-4, OE-34 and OE-55 were about 38.10%, 34.52% and 34.52%, which were significantly lower than those infected by WT (63.10%) and VT (72.62%) at 7 days after the topical inoculation (Figure 4A). Additionally, the LT_50_s of OE-4 (6.41 ± 1.44 d), OE-34 (6.25 ± 0.63 d) and OE-55 (6.27 ± 0.80 d) were significantly shorter than those of WT (8.20 ± 0.16 d) and VT (8.59 ± 0.34 d, *p* < 0.05; Figure 4B).

The fungal growth rates in vivo were determined using quantitative PCR (qPCR). The results showed that the DNA concentrations of OE-4 and OE-55 in locust hemolymph were 79.09 ± 0.85 pg/μL and 76.23 ± 4.41 pg/μL at 3 days after inoculation, which were significantly higher than 48.09 ± 7.20 pg/μL for WT, respectively (*p* < 0.05; Figure 4C). After topical inoculation for 5 days, the DNA concentrations in locust hemolymph were 181.02 ± 23.38 pg/μL and 192.58 ± 23.02 pg/μL for OE-4 and OE-55, which were significantly higher than WT (108.94 ± 18.06 pg/μL) and VT (107.44 ± 16.56 pg/μL, *p* < 0.05; Figure 4C), respectively. Consistently, the hyphal bodies were increased in the locusts infected by the *MaSom1*-overexpression strains on the 3rd and 5th day after inoculation (Figure 4D).

The assays of conidial germination and appressorium formation were performed on the hind wings of the locusts. The conidial germination rates of the OE-4 (49.33 ± 2.33%), OE-34 (64.00 ± 5.00%) and OE-55 (51.00 ± 4.00%) strains were significantly higher than those of WT (32.67 ± 4.67%) and VT (31.50 ± 2.50%, *p* < 0.05; Figure 5A) at 8 h after inoculation. The conidial germination rates of WT (93.00 ± 1.00%) and VT (91.50 ± 0.50%) strains were closed to those of OE-4 (97.67±1.33%), OE-34 (96.33 ± 2.33%) and OE-55 (94.00 ± 3.00%, *p* > 0.05; Figure 5A) strains at 36 h after inoculation. The mean 50% germination times (GT_50_s) for WT (11.10 ± 0.92 h) and VT (12.76 ± 0.28 h) were significantly longer than those of OE-4 (9.50 ± 0.60 h), OE-34 (6.65± 0.73 h), and OE-55 (7.30 ± 0.63 h, *p* < 0.05; Figure 5B). At 8 h after inoculation, no appressorium formation was found in the control strains, while about 8% of the germinated conidia had formed typical appressoria in the *MaSom1*-overexpression strains (Figure 5C,F). After incubation for 12 h, the appressorium formation rates of the *MaSom1*-overexpression strains were more than 44.00%, whereas the appressorium formation rates in the control strains were about 35.33% (Figure 5C). At this time, the longer germination tubes were seen in the *MaSom1*-overexpression strains (Figure 5F). Until 28 h after incubation, the appressorium formation rates (~84%) of all fungal strains showed no difference (Figure 5C). The adhesion ratio of *MaSom1*-overexpression strains (87.80 ± 1.28% for OE-4, 84.55 ± 2.12% for OE-34, 82.62 ± 2.29% for OE-55) to locust hind wings is significantly higher than that in WT (74.42 ± 1.95%) and VT (74.22 ± 8.14%) after inoculation for 8 h (Figure 5D). The relative expression levels of *Metarhizium* perilipin-like protein gene *MaMPL1*, the neutral trehalase gene *MaNTH1* and the adhesin protein gene *MaMad2* during appressorium formation were increased in the three *MaSom1*-overexpression strains (Figure 5E), indicating that the adhesion ability and glycerol biosynthesis might be much more effective in OE-4, OE-34, OE-55 than in WT and VT. Taken together, the augmented virulence of the *MaSom1*-overexpression strains is partly due to the enhanced adhesion and penetration of the appressoria.

## 3. Discussion

Entomopathogenic fungi are important environmentally natural factors for controlling the populations of insect pests [35]. Unlike bacteria and viruses, fungi can infect insects by penetration of the host cuticle directly [36]. Entomopathogenic fungi exhibit huge application potential, because they could infect not only chewing pests but also sucking pests, and even pests at no feeding stage such as pupae or eggs [37]. However, the disadvantages, such as poor efficacy, environmental instability and high cost, still slow their wide application [5,8,9]. Genetic engineering has supplied many approaches for fungal improvement to overcome their shortcomings via manipulation the genes related to fungal stress tolerances and pathogenesis, insect development and immunity [15,16,38,39,40]. However, previous fungal improvements are only related to single or partial biocontrol properties.

Transcription factors usually simultaneously regulate multiple target genes related to fungal development, virulence and stress tolerances [2]. Therefore, some core transcription factors in key regulator pathways may act as important potential candidates for genetic improvements of entomopathogenic fungi. In this work, overexpression of *MaSom1*, a key transcription factor gene in the cAMP/PKA pathway, enhanced the biocontrol potential by increasing stress tolerance, promoting conidiation, and augmenting the virulence of *M. acridum*.

The IT_50_s of the *MaSom1*-overexpression strains were increased by 155.11% for heat shock and 66.77% for UV-B when compared with WT, which exhibit a comparable tendency when introducing 1,3,8-trihydroxynapthalnene reductase into *M. anisopliae*, with an increase in IT_50_s of 32.44% for UV-B radiation (23.4 mJ cm^−2^) and 137.69% for heat shock [15]. The higher expression levels of *MaUve1*, *MaUbi1*, and *MaSsb1* in the *MaSom1*-overexpression strains indicated that the overexpression of *MaSom1* resulted in improvements in fungal tolerance to UV-B and heat shock via enhancing the expression of genes involved in fungal stress tolerances. Unlike how disruption of *Mamhk1* in *M. roberstsii* increased the final conidial yield by up to 50.00% in 1/4 SDAY medium [20], the *MaSom1*-overexpression strains showed no significant increase in the final conidial yield when compared with the control strains on the 15th day, which may be due to the nutritional restriction after the 9th day for fungal growth, but advanced conidiation was presented with an increased conidial yield by up to 288.95% on the 3rd day. The acceleration of fungal conidiation will shorten the time for producing conidia by fermentation, which is very important for promoting the industrial production of mycoinsecticides [41].

Fungal virulence has been increased via genetic engineering by improving the progress of host cuticle penetration, fungal immune escape, fungal growth in vivo, or introducing toxins or insect molecules to affect insect behaviors, previously [42,43]. Overexpression of cuticle-degrading protease *MaPr1A* in *M. anisopliae* enhanced fungal virulence with an LT_50_ (93 h) about 22.50% shorter than that of WT (120 h) when infected the *Manduca sexta* larvae [23]. Overexpression of the insect serpin gene *Spn43Ac* decreased the LT_50_s of *B. bassiana* by 16.00% to 24.00% via inhibiting the insect immune system to promote hyphal body growth in host hemolymph [29]. Introducing the *Aedes aegypti* trypsin-modulating oostatic factor gene into *B. bassiana* significantly augmented the fungal virulence, and the LT_50_s of the transgenic strain decreased by about 24.51% and 35.38% after treating the adults and larvae *A. aegypti*, respectively [28]. In this article, after topical inoculation on the locust, the LT_50_s of the *MaSom1*-overexpression strains were decreased by 21.83% to 23.78% in contrast with WT, which is comparable to the previous fungal improvements. The *MaSom1*-overexpression strains exhibited increased adhesion ability, glycerol biosynthesis of appressoria, and hyphal body growth in host hemolymph, indicating *MaSom1*-overexpression augmented the fungal virulence due to not only enhanced adhesion and penetration with advanced appressorium formation but also accelerated fungal growth in locust hemolymph. It is indicated that when compared with previous research, a diverse capability of adaptation to the natural environment has been significantly improved in the *MaSom1*-overexpression strains. The results also provide excellent clues to the huge potential of key transcription factors in fungal improvements through genetic engineering.

The various adversities are the main factors that limited the effectiveness of the mycoinsecticides [2,12,14,44,45,46]. The adaptation mechanisms of these adversities are complex, as they involve interactions with many genes in multiple pathways and transcription regulate network [13,47,48,49,50]. Here, we show that when a key transcription factor of an important regulatory pathway was expressed constitutively, fungal strains with improvements in multiple respects were obtained. This provides a new strategy through changing the regulation progress of some key transcription factors by genetic engineering to improve the biocontrol potential of entomopathogenic fungi, which exhibits an important practical significance.

Furthermore, we should realize that many of these ideas are derived from the rapid development of biological engineering technology. An increased number of transgenic strains are being obtained, and all of these results may provide more insights for fungal improvements. This exiting research has shown that, upon introduction of the transcription factors that regulate two different metabolic pathways into rice, disease resistance and salt tolerance were both significantly improved [51], therefore the overexpression of several selected genes in fungi is also worth considering. In addition, combining multiple physical and biological mechanisms to work together may also be effective. Combining the engineering strains with UV absorbers and humidity stabilizers will strongly increase the potential for stress tolerance [52].

Understanding of fungal development and pathogenesis, and the detailed knowledge from the further research on entomopathogenic fungi, will lead to the creation of other, new, rational design strategies for improving the effectiveness of mycoinsecticides [53]. Accompanied with the rapid development of biological engineering technology, we may predict the prosperous application of the engineering entomopathogenic fungi.

## 4. Materials and Methods

### 4.1. Microbial Strains and Maintenance

All constructed strains in our study were generated from *M. acridum* CQMa102 which is conserved at China General Microbiological Culture Collection Center (WT, No. 0877), and cultured in 1/4 SDAY medium (one-quarter-strength Sabouraud dextrose agar medium which consists of 1% dextrose, 0.50% yeast extract, 0.25% mycological peptone and 2% agar, *w*/*v*) for 15 days at 28 °C. For testing the fungal cell wall integrity, fungal spot assays were conducted on 1/4 SDAY, 1/4 SDAY supplemented with 50 μg/mL CFW and 500 μg/mL CR at 28 °C for 5 days according to a previous method [34]. The diameters of the fungal colonies were measured to calculate the relative growth inhibition. For DNA manipulations and fungal transformations, the *Escherichia coli* DH5α (TransGen Biotech, Beijing, China) and *Agrobacterium tumefaciens* AGL-1 (TransGen Biotech, Beijing, China) growth on Luria–Bertani (LB) broth were selected at 37 °C and 28 °C, respectively.

### 4.2. Vector Constructions and Fungal Transformations

The *MaSom1* coding sequence (2762 bp) was amplified according to the CQMa102 genomic DNA template with the primer pair MaSom1-F and MaSom1-R (Appendix A) and inserted into the pK2-P*_gpdA_*-egfp vector, which contained the *PgpdA* promoter and the terminator TtrpC from *A. nidulans*, with *Xba*I/*Bam*HI digested to produce the *MaSom1*-overexpression vector pK2-P*_gpdA_*-MaSom1::egfp. The pK2-P*_gpdA_*-egfp and pK2-P*_gpdA_*-MaSom1::egfp vectors were sequenced and mobilized into *A. tumefaciens* AGL-1 for fungal transformation [54]. The empty vector strains and the *MaSom1*-overexpression strains were screened on Czapek-dox agar medium (QDRS BIOTEC, Qingdao, China) with 20 μg/mL chlorimuron ethyl (Sigma, Bellefonte, PA, USA). The putative *MaSom1*-overexpression and the empty vector transformants were screened by PCR with the primer pairs of MaSom1-RF/EGFP-VR and PgpdA-VF/EGFP-VR, respectively (Appendix A).

### 4.3. Validation of MaSom1 Overexpression Strains

*MaSom1*-overexpression and VT strains were further selected via fluorescent signal detection from the putative transformants, as enhanced green fluorescent protein (egfp) was fused. The mature conidia were harvested from fungal cultures grown on 1/4 SDAY agar plates for 15 days. Fungal hyphae were harvested from 1/4 SDAY agar plates which spread evenly with 100 µL conidial suspension (5 × 10^6^ conidia/mL) and cultured at 28 °C for 18 h. All the samples were washed with 1% PBS three times. The conidia and hyphae were stained with the nuclear dye Hoechst 33258, and the subcellular location was detected by the position of blue nuclear fluorescence and egfp fluorescence. The green and blue fluorescence in fungal strains were detected by the fluorescence microscopy (Nikon Y-TV55, Tokyo, Japan) with an excitation light for 488 nm and 352 nm, respectively [34]. After that, three *MaSom1*-overexpression strains and one VT strain which exhibited higher fluorescence in conidia, designated as OE-4, OE-34, OE-55 and VT, were selected and verified. The mature conidia from 1/4 SDAY plates were washed with 1% PBS and harvested to detect the expression levels of *MaSom1* in WT, OE-4, OE-34, OE-55 and VT by qRT-PCR with the primer pair of MaSom1-QF/MaSom1-QR (Appendix A). Total RNA extraction, cDNA reverse transcription, and qRT-PCR were performed according to the existing method [55]. The glyceraldehyde 3-phosphate dehydrogenase gene *gpdh* (GenBank accession No. XM_007817733.1) acted as an internal control, which tested the primer pair of Gapdh-QF/Gapdh-QR (Appendix A). All qRT-PCR amplifications were performed using three biological replicates of RNA samples. Southern blotting was performed to further validate the *MaSom1*-overexpression and the empty vector strains. All the genomic DNA was digested with *Eco*RV, transferred onto an Immobilon-Ny + transfer membrane (Millipore, Bedford, MA, USA), and then detected with the Detection Starter Kit I (Roche, Mannheim, Germany). A part of the *sur* gene (319 bp) was used amplify the probe1 (*sur*-PF/*sur*-PR, Appendix A) and was labelled with Digoxigenin HighPrime.

### 4.4. Insect Bioassays

The *Locusta*
*migratoria manilensis* fifth-instar nymphs reared by our laboratory were used for bioassays with topical inoculation according to the existing method [56]. Five microliters of paraffin oil conidial suspensions (1 × 10^7^ conidia/mL) of all the tested fungal strains were dipped onto the head–thorax junction of the fifth-instar nymphs locust. The control locusts were treated with an equal volume of paraffin oil. All the locusts were reared at 25 °C after topical inoculation. For each fungal strain, 3 × 30 locusts were used. The mortality of locusts was recorded for the 12 h interval.

The rates of the conidial germination and the appressorium formation were examined from 4 to 36 h with a 4 h interval on locust hind wings by the method described previously [57]. Additionally, microscopic observation of the appressorium morphology was performed at 8, 12 and 24 h after inoculation. The adherence assays were performed using a method as described previously [58]. The conidial suspensions were inoculated on locust hinds to induce appressorium formation at 28 °C for 20 h. The total RNA extracted from the cultures was used to detect the relative expression levels of the genes related to adhesion and penetration. The hyphal bodies were harvested from the hemolymph of the fifth-instar nymphs at 3 and 5 days after topical inoculation. The fungal growth rates in insect hemolymph in vivo were determined by quantifying the *M. acridum* genomic DNA via qPCR with the primer pair ITS-F and ITS-R (Appendix A) [59]. Briefly, 300 µL hemolymph (30 µL hemolymph per locust) was extracted from the treated locust, then centrifuged for 3 min at 12,400× *g* and rinsed with sterile water for total DNA extraction. A standard curve was made using the series of *M. acridum* genomic DNA with gradient concentrations of 65.6 pg/µL, 131.2 pg/µL, 263 pg/µL, 525 pg/µL, 1050 pg/µL, 2100 pg/µL, 4200 pg/µL, 8400 pg/µL. All experiments were performed in triplicate.

### 4.5. Measurements of Conidial Yield and Stress Tolerances

Conidial yield was determined according to the method described previously [34]. In brief, an aliquot of 2 μL conidial suspension (1 × 10^6^ conidia/mL) was inoculated in 24-well plates with 1 mL 1/4 SDAY medium in each well. The concentrations of the conidial suspensions were calculated with a hemocytometer by harvesting conidia from each well into 1 mL 0.1% Tween-80 from 3 to 15 days with a 2-day interval. The mycelial morphology was observed and photographed at 12, 15, 18, and 21 h after inoculation with the light microscopy, Leica microscope DM750 (Leica Microsystems, Wetzlar, Germany). The fungal tolerances to heat shock and UV-B irradiation were examined as described previously [60]. Simply, the conidial germination rates of fungal strains were calculated after treatment and then cultured for 20 h on 1/4 SDAY medium. Additionally, 50 μL of conidial suspensions (1 × 10^7^ conidia/mL) were exposed for 0.0, 2.0, 4.0, 6.0, and 8.0 h at 45.0 °C for heat shock treatment, and the evenly distributed conidia were exposed to UV-B irradiance of 1350.0 m W/m^2^ for 0.0, 1.5, 3.0, 4.5, 6.0, 7.5 and 9.0 h. All the strains were cultured at 28 °C and all experiments were repeated in triplicate.

### 4.6. Statistical Analysis

The Data Processing System program (DPS) was used to estimate the GT_50_, the IT_50_ and the LT_50_ [61]. Statistical significances of all the results were analyzed with SPSS 16.0 (SPSS Inc., Chicago, IL, USA) by one-way analysis of variance (ANOVA). Tukey’s honestly significant difference test was used to separate means at α = 0.05. All experiments were performed at least three times.

## Figures and Tables

**Figure 1 jof-08-00105-f001:**
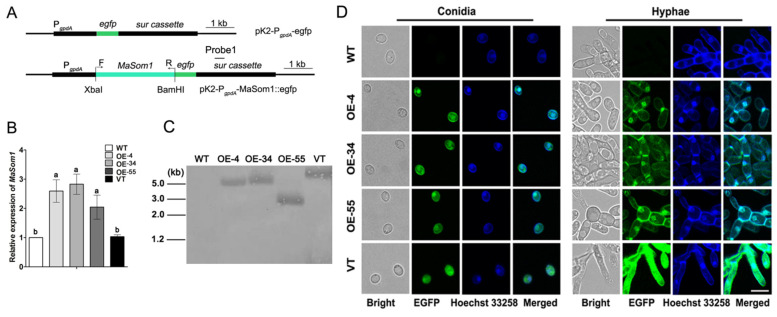
The overexpression of *MaSom1* in *Metarhizium acridum*. (**A**). Schematic illustration of the *MaSom1*-overexpression in *M. acridum*. (**B**). Relative expression of *MaSom1* in fungal strains. Different lowercase letters on bars denote significant differences between samples for *p* < 0.05. (**C**). Validation of *MaSom1*-overexpression strains using Southern blotting. The genomic DNA was cut with *Eco*RV. Probe1 was amplified from plasmid DNA by PCR using primer pair *sur*-PF and *sur*-PR (Appendix A). (**D**). The fluorescence observation. Bar = 5.0 µm.

**Figure 2 jof-08-00105-f002:**
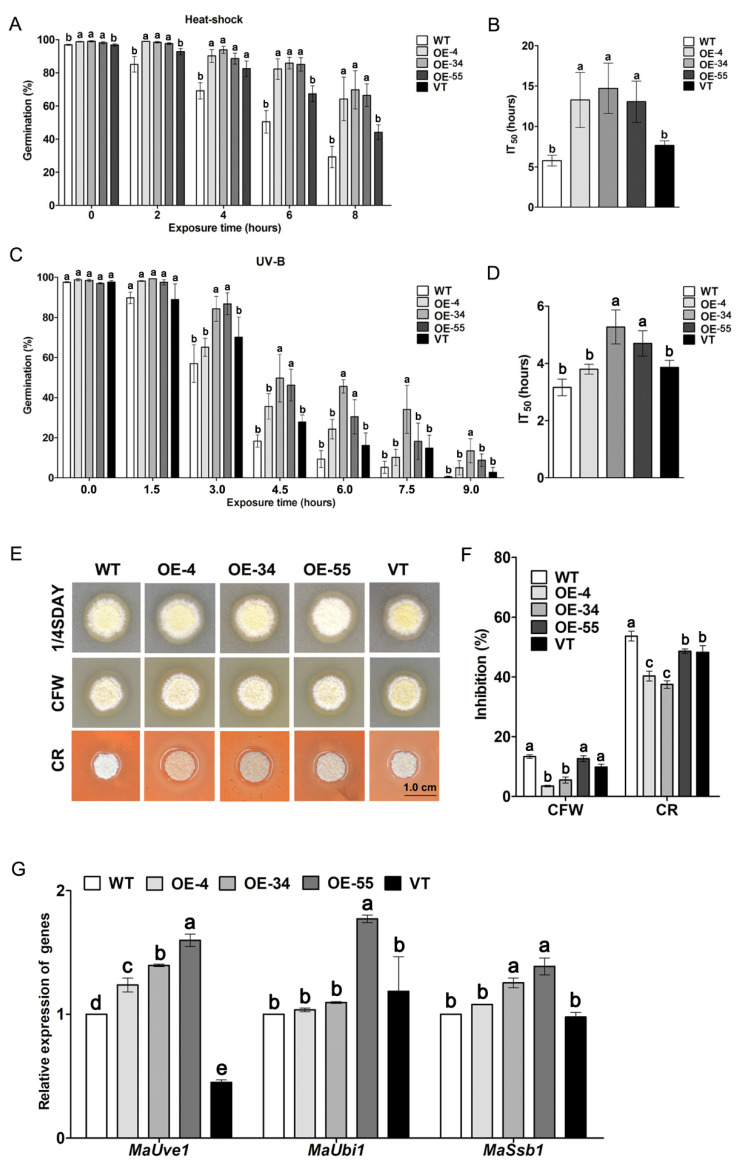
The tolerances of heat shock and UV-B irradiation were increased in the *MaSom1*-overexpression strains. (**A**). The conidial germination rates of fungal strains after heat shock. (**B**). The IT_50_s of fungal strains after the heat shock. (**C**). The conidial germination rates of fungal strains after the UV-B irradiation. (**D**). The IT_50_s of fungal strains after the UV-B irradiation. (**E**). Colony morphology on 1/4 SDAY medium and 1/4 SDAY medium supplemented with calcofluor white (CFW, 50 mg/L) and Congo red (CR, 500 mg/L). (**F**). Fungal growth inhibition rates on 1/4 SDAY medium with CFW or CR. (**G**). Relative expression of genes related to protection against UV-B and heat shock in fungi. *MaUve1* (MAC_07337), a gene encoding a UV endonuclease; *MaUbi1* (MAC_01946), a gene encoding a ubiquitin; *MaSsb1* (MAC_07411), a heat shock protein 70 gene. Different lowercase letters on bars denote significant differences between samples for *p* < 0.05.

**Figure 3 jof-08-00105-f003:**
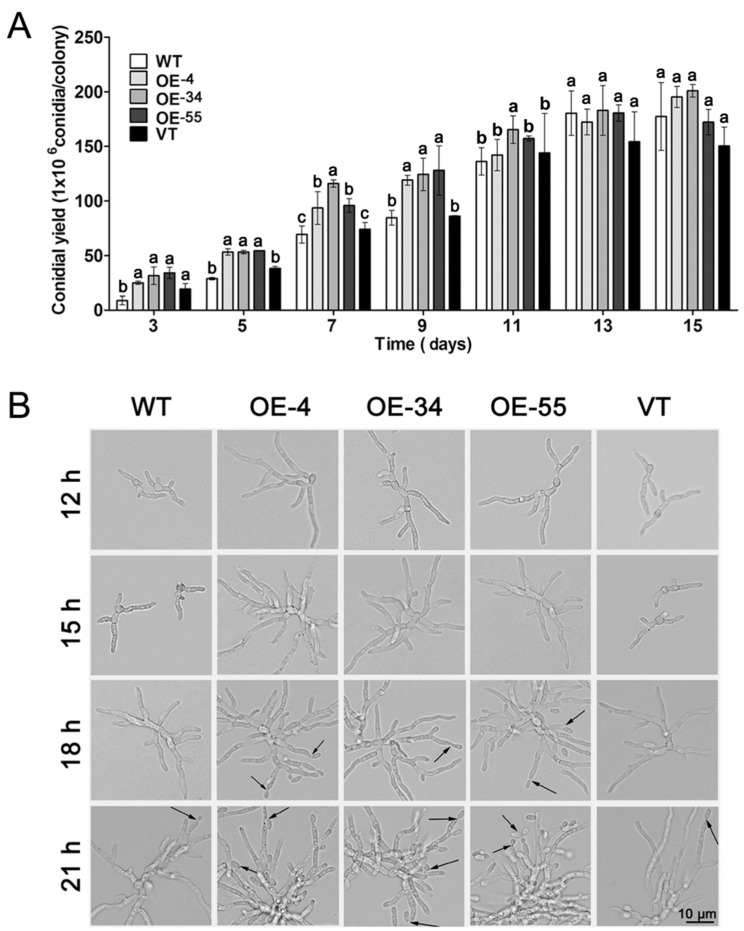
The conidiation of the *MaSom1*-overexpression strains were accelerated. (**A**). The conidial yield of fungal strains grown on 1/4 SDAY medium. (**B**). The microscopic observation of hyphal growth and conidiation of fungal strains. Different lowercase letters denote significant differences between the fungal strains for *p* < 0.05. The black arrows represent the conidia. Bar = 10.0 µm.

**Figure 4 jof-08-00105-f004:**
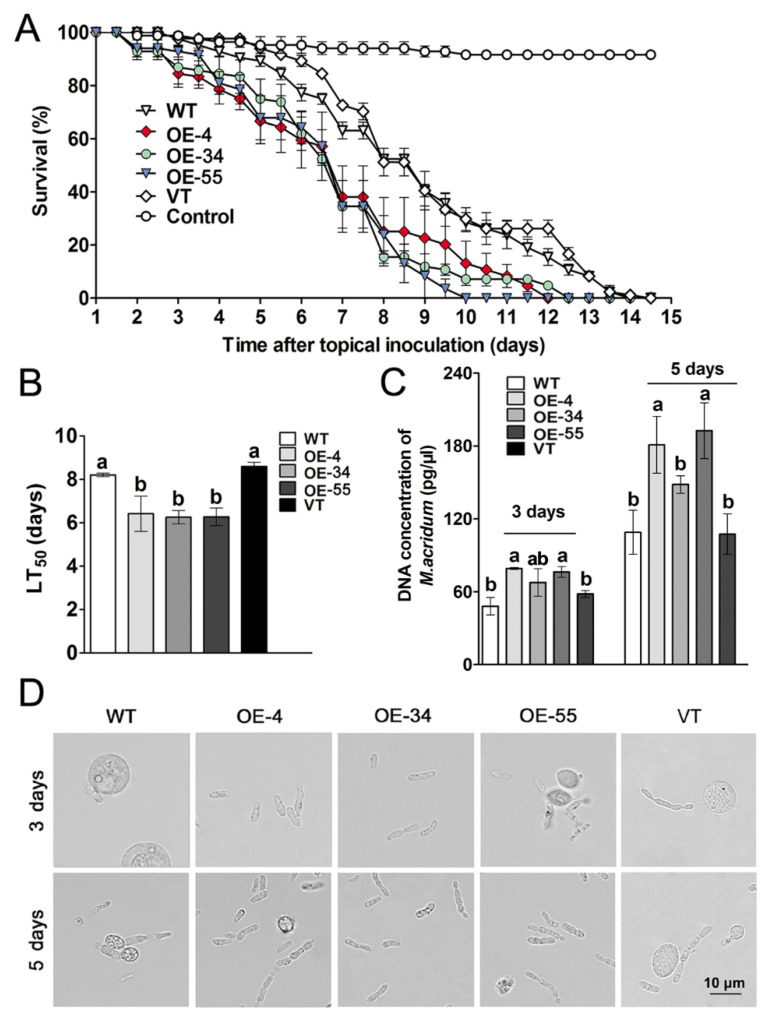
The virulence of the *MaSom1*-overexpression strains was increased. (**A**). The locust survival rates after topical inoculation of paraffin oil conidial suspension for WT, OE-4, OE-34, OE-55 and VT strains, respectively. (**B**). The LT_50_s of WT, OE-4, OE-34, OE-55 and VT strains. (**C**). The DNA concentrations of the WT, *MaSom1*-overexpression and VT strains in locust hemolymph in vivo after topical inoculation for 3 and 5 days. (**D**). The microscopic observation of the hyphal bodies of fungal strains in locust hemolymph in vivo. Bar =10.0 µm. Different lowercase letters on bars denote significant differences between samples for *p* < 0.05.

**Figure 5 jof-08-00105-f005:**
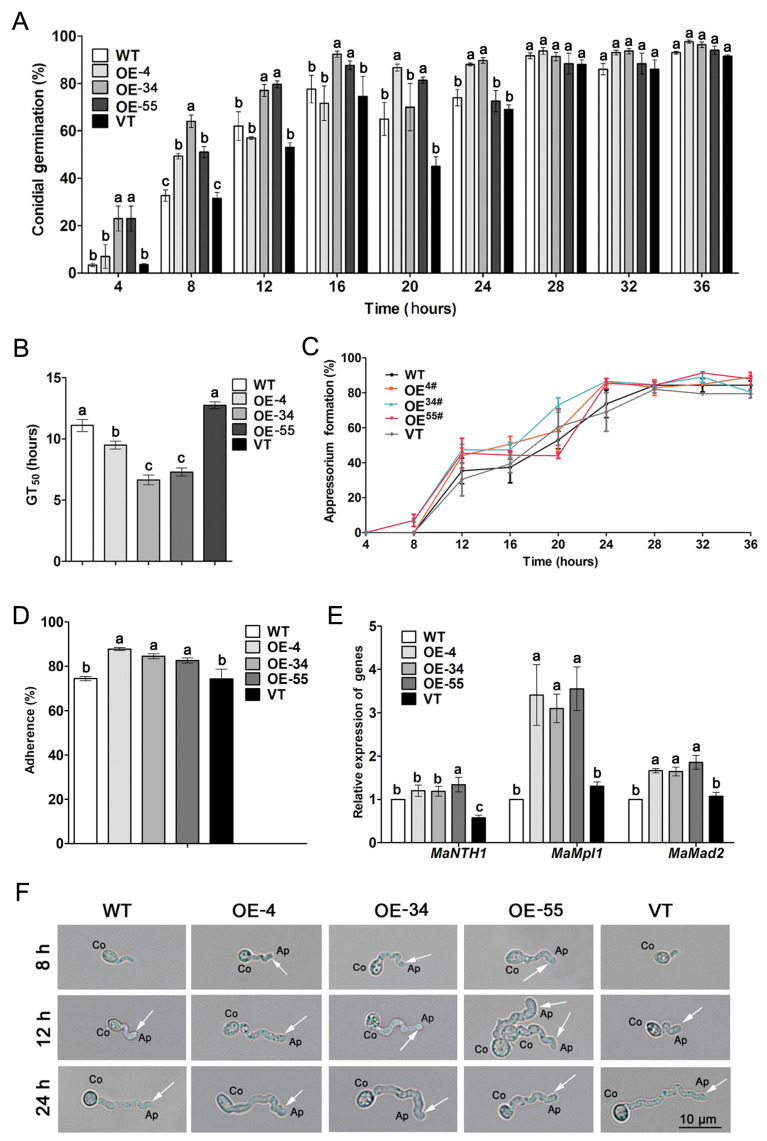
The conidial germination and appressorium formation on locust hind wings were advanced in the *MaSom1*-overexpression strains. (**A**). The conidial germination rates of the fungal strains on locust hind wings. (**B**). The GT_50_s on locust hind wings of fungal strains. (**C**). Appressorium formation of fungal strains on locust wings. (**D**). The adherence ratio of *M. acridum* conidia to locust hind wings. (**E**). Relative expression of genes involved in glycerol biosynthetic pathway. *MaMPL1* (XM_007813754.1), a gene encoding a perilipin-like protein in *M. acridum*; *MaNTH1* (XM_007814339.1), a gene encoding neutral trehalase *in M. acridum*; *MaMad2* (XM_007809102.1), a gene encoding an adhesin protein in *M. acridum*. (**F**). The morphology of appressoria from different fungal strains. The white arrows indicate the typical appressoria. Co: conidium, AP: appressorium. Different lowercase letters on bars denote significant differences between the fungal strains for *p* < 0.05.

## Data Availability

All relevant data are within the paper and its Appendix A.

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
