# Peer review of "Enhancing the Biocontrol Potential of the Entomopathogenic Fungus in Multiple Respects via the Overexpression of a Transcription Factor Gene *MaSom1"

_jof, 2022, doi:10.3390/jof8020105_

Round 1
Reviewer 1 Report
In this article, the overexpression of the transcription factor MaSom1 in the entomopathogenic fungus M. acridum is evaluated. Three strains that overexpress the MaSom1 gene are analyzed and selected from 63 correct fungal transformants.
The authors show an interesting article with the experiments and results. My only observation concerns that is not shown, that the integrating the MaSom1 gene overexpression cassette is not affecting some other gene in the fungus.
I believe it is important to show that in the three overexpressants used, the insertion of the overexpression cassette was in different loci of the fungal genome. This because if the insertion in the three transformants corresponds to the same locus, it is possible that the observed effects are because of the alteration of some gene by the integration PgpdA-MaSom1:: egfp cassette. I based the observation on the results that show that the VT strain transformed with the PgpdA-egfp cassette also has an increase in the tolerances of heat shock and UV-B irradiation.
The verification of the integrations are in different loci could be shown by southern blot or some genome walking strategy to know the flanking region to the integration cassette.
Reviewer 2 Report
The manuscript “Comprehensively Enhancing the Biocontrol Potential of the Entomopathogenic Fungus via the Overexpression of a Transcription Factor Gene MaSom1” described the overexpression of MaSom1, which in turn enhances the fungal virulence, resistance to environmental stressors, as well as enhances the germination and conidiation rate. Although the manuscript brings wonderful results and the generated transgenic strains seem promising, some aspects should be improved. Firstly, the English quality must be improved, avoiding the overuse of some words as “comprehensive”.
Major points:
1. It is quite interesting that the authors did not use a construct to overexpress MaSom1 without the GFP tag. In the end, it does not matter much since the boost in the virulence and several other aspects are the focus of the manuscript. However, biologically, would be interesting to get an explanation from the authors. Did they try it? Better results would be obtained?
2. The authors described that 63 mutants were obtained through ATMT, but only 3 mutants were explored. The other 60 mutants also displayed a similar phenotype when compared to the chosen ones? The authors did not explain in the Material and Methods section how do they selected the OE-4, OE-34, and OE-55.
3. How many copies of the MaSom1 overexpression cassette were integrated into the M. acridum genome? Usually, we tend to believe that ATMT only generates one integration event per mutant, but that is not always true. Would be interesting to explore by southern blotting how many copies were integrated into OE-4, OE-34, and OE-55 genomes. Furthermore, where do these copies land? The aleatory integration of the expression cassette can lead to several mutations which, in turn, can affect the observed phenotype. Would also be interesting to explore if secondary mutations were not generated and if these secondary mutations would not have an influence on the observed phenotype. If the authors can prove that the overexpression cassette landed in different regions of OE-4, OE-34, and OE-55 genomes that would be enough for me.
Reviewer 3 Report
Dear Editor
The manuscript by Du and coworkers describes the functional enhancement of specific trais of Metarhizium acridum related to stress tolerance and virulence due to theoverexpression of a transcription factor encoding gene. After construction of strains that harbor a cassete to overexpress MaSom1, a transcription factor associated with the cAMP/PKA pathway, the authors found that such strains are more tolerante to heat shock and UV radiation compared to WT, presented higher germinations and comicial yield, as well as increased virulence to locusts. This is a clear follow up of the previous publication of the group, which described the generation of null mutants of MaSom1. The results presented are solid and the manuscript is well written. However, some conclusions draw by the authors are not fully supported by the presented data. These are summarized bellow:
1 - While the results presented clearly show that the over expression of MaSom1 drives intense phenotypic alterations, the molecular mechanisms are not explored. It is not clear from the manuscript if the constructions were made to be inserted in the homologous locus or to be integrated ectopically into the genome. In this way, even with three distinct mutants, the observed phenotypes could be derived from ectopic integration of the cassetes. In order to rule this out, the expression of MaSom1 putative targets should be evaluated by qRT-PCR. For instance, it is expected that some chaperones (HSP70) should be increased in the over expression mutants. Potential targets could be inferred form the orthologs from Verticillium dahliae (https://doi.org/10.1111/nph.15514) or Aspergillus fumigatus (https://doi.org/10.1371/journal.ppat.1005205 and https://doi.org/10.1128/mBio.02329-20).
2 - The data of nuclear localization in figure 1c must be confirmed by DAPI staining and colocalization analysis.
3 - As previously described, M. acridum cells lacking MaSom1 displayed decreased expression of adhesins and adhesion related genes. What would be the expression of such genes in the over expression strains and how would it correlated to increased virulence? Moreover, what would be the adhesion of such strains to host (please see the experiment in https://doi.org/10.1128/EC.00409-06)
4 - Would such Overexpression strains also displays increased tolerance to cell wall stressors?
5 - despite no differences in appressorium formation, it is possible that such apressaria are more efficient. This could be evaluated by the measurement of hydrolytic enzymes activity, as chitinases and proteases.
Minor points
6 - The sentence "Though fungal genetic improvements have been performed through different strategies, these improvements of single or partly biocontrol properties are not satisfied to overcome the shortcomings of mycoinsecticides in field. " in abstract is not clear and needs alteration to enhance it understanding.
7 - Abstract, the word necessary in the sentence "Therefore, comprehensively enhancing the biocontrol potential of entomopathogenic fungi by genetic engineering is necessary." Is too strong. Please replace by "desirable".
8 - Introduction. Rephrase "Disruption Mamhk1 (a group III histidine kinase gene)" to "Disruption of Mamhk1 (a group III histidine kinase gene)"
9 - Introduction. What are the 63 correct fungal transformants? How are they evaluated for its correctness? Maybe other terms should be more adequate to be placed in the introduction, as putative transformants or transformants continuing the expression cassette.
10 - item 2.1 the NCBI accession codes refer to protein sequences, which can't be cloned. Please specify the nucleic acids accession codes. Moreover, please specify if the CDS (from cDNA or from genomic DNA) was amplified. Finally, describe the terminator used in the construction
11 - Item 2.2. What IT50 stands for? Please describe the abbreviation in its first use
12 - Figure 4C. At 3 days post infection, it is very confusing that OE4 and OE55 strains differs from OE34 by ANOVA tests, based on the media and SDs. Please revise such data.
13 - Please quantify baslatospores from figure 4D to correlate with data from figure 3C. Whithout any measurements, there no new information in Fig 4d.
14 - Discussion - the sentence "Transcription factors usually simultaneously regulate multiple target genes related to fungal development, virulence and stress tolerances by combining with corresponding factors [2]. " needs polishing. At this state is confusing. What is "by combining with corresponding factors?
15 - In the last paragraph os the discussion, capitalize the first letter of understanding
16 - Are primers ITS-F and ITS-R specific to Metarhizium? As a widely used region to be amplified and sequenced for species identification, it could amplify also the potential normal mycobiome of the host. Please discuss on this.
Round 2
Reviewer 2 Report
The authors fulfilled all my questions. I have three comments:
1 - I do not like that the authors used lowercase and capital letters to display significant statical differences. The authors could modify the figures to present only lowercase or capital letters.
2 - Legend in Figure 5: M. acridum was miswriteed.
3 - In the experiment of DNA concentration in the hemolymph if the ITS barcode sequence was used (i. e., a sequence which can be amplified from all fungal species) a control of "clean hemolymph", without M. acridum infection, should be used.
Reviewer 3 Report
Dear Editor,
The manuscript was significantly improved regarding the first version submitted. I would like to suggest only a minor revision, which do not require further verification by reviewers. In the begging of results section, change XM_007810626.1 by GeneID code MAC_02477. The XM_007810626.1 refers to a transcript entry in NCBI. As the authors stated that the genomic DNA was the source for amplification of SOM1, it would be better to properly represent the genomic sequence.
With this minor modification, the manuscript will meet the publication criteria of JOF.
